# Use of Tremella as Fat Substitute for the Enhancement of Physicochemical and Sensory Profiles of Pork Sausage

**DOI:** 10.3390/foods10092167

**Published:** 2021-09-13

**Authors:** Hewen Hu, Yue Li, Long Zhang, Huajie Tu, Xinyu Wang, Lili Ren, Siqi Dai, Liyan Wang

**Affiliations:** 1College of Food Science and Engineering, Jilin Agricultural University, 2888 Xincheng Street, Changchun 130118, China; huhewen@jlau.edu.cn (H.H.); liyue970523@163.com (Y.L.); 20200704@mails.jlau.edu.cn (L.Z.); 20200816@mails.jlau.edu.cn (H.T.); 20200828@mails.jlau.edu.cn (X.W.); daisiqi1232021@163.com (S.D.); 2College of Humanities (College of Home Economics), Jilin Agricultural University, 2888 Xincheng Street, Changchun 130118, China; 3Key Laboratory of Bionic Engineering (Ministry of Education), College of Biological and Agricultural Engineering, Jilin University, 5988 Renmin Street, Changchun 130022, China; liliren@jlu.edu.cn

**Keywords:** tremella, fat substitute, sausage

## Abstract

Pork fat in sausage was replaced by tremella at different proportions during the process, and the physicochemical and sensory profiles of pork sausage were evaluated. Five recipes with the replacement proportion of 0%, 25%, 50%, 75%, and 100% tremella were manufactured, and their proximate compositions, water activities, textures, colors, water holding capacities (WHC), and amino acid compositions were investigated. The results showed that the protein, ash and moisture content, lightness, redness, and WHC of pork sausages were increased (*p* < 0.05), and textural profile analysis (TPA) and sensory quality of the sausage were improved (*p* < 0.05). In addition, the content of essential amino acids (lysine and isoleucine) and the non-essential amino acids (proline and tyrosine) of sausages were increased (*p* < 0.05). The sausage had the best sensory performance when the replacement ratio of tremella was 75%. These results indicated that replacing fat with tremella could be a valid way to obtain nutritional and healthy sausage.

## 1. Introduction

Sausages are favored all over the world for their high nutritional value and delicious taste [1], and have important economic significance for the meat-packing industry. The main ingredients of sausages are pork lean meat, pork fat, isolated soy protein, and starch. In general, sausages are high in fat, exceeding 20%. However, the world health organization (WHO) recommends reducing fat intake [2], since excessive fat intake could lead to coronary heart disease, cardiovascular disease, hypertension, and other diseases [3,4,5]. Due to the popularity of sausages by consumers, it is necessary to develop low-fat sausages to reduce health problems.

Some meat technicians used plant additives (soy protein isolate and cellulose nanofiber complex gels) as fat substitutes to produce low-fat meat products [6], but consumers often shun these plant additives which were obtained through chemical or transgenic pathways. Nowadays, natural and healthy food has become a research hotspot. Zhu et al. [7] improved the fatty acid composition and sensory quality of sausages by using Jerusalem artichoke powder and olive oil to replace fat. Yinyu et al. [8] added regenerated cellulose fiber to the fat-reduced emulsified sausage, which could effectively reduce the fat content without affecting the quality and sensory characteristics. In addition, Utama et al. [9] found that the pre-emulsified perilla-canola oil could improve the hardness and maintain the acceptable appearance, flavor, and overall impression of the sausage. Najjar et al. [10] added emulsified rapeseed oil and protein-based fat substitute to the beef filling, which could increase the hardness of the beef filling and reduce cooking loss.

In addition, camellia oil gel [11], chicory root powder [12], transesterified palm kernel oil [13], hydroxypropyl methylcellulose oil gel [14], hazelnuts [15], rye bran fiber and collagen [16], plant powder and konjac gel [17], and Chitosan and gold flax seed powder [18] were also used as fat substitutes in meat products.

Tremella is rich in protein and trace elements, so it could improve the detoxification ability of liver, protect the liver, and could be able to enhance the body’s anti-tumor immunity [19]. Tremella extracts possess capacities of anti-fatigue and anti-hypoxia [20]. Tremella is crystal clear, juicy and sticky, and resembles pig fat in appearance, showing good properties as a fat substitute. Therefore, tremella could be a good strategy to be obtained in the development of low-fat sausage. The objective of this research was to manufacture sausages added with tremella and investigate the effects of tremella as a fat substitute for low-fat sausages by measuring proximate component, color, water activity, pH, cooking loss, water holding capacity, textural profile analysis, free amino acids, and sensory evaluations.

## 2. Materials and Methods

### 2.1. Materials

Fresh pork was purchased from Jilin Huazheng Agriculture and Animal Husbandry Development Co., Ltd. (Changchun, Jilin, China). Tremella (basswood tremella, protein: 14%; fat: 3%, carbohydrate: 16%) was obtained from Heilongjiang Lvzhiyuan Agricultural and Sideline Products Co., Ltd. (Suihua, Heilongjiang, China). All of the additives were supplied by Sichuan Jinshan Pharmaceutical Co., Ltd. (Meishan, Sichuan, China) and all the chemical reagents were supplied by Beijing Beihua Co., Ltd. (Beijing, China).

### 2.2. Sausage Formulation and Processing

Dried tremella (100 g) was soaked in warm water at 40 °C for 30 min and cut into slices with a width of 0.5 cm. The lean meat and fat were ground through a 15 mm plate (TJ12, Panyu Liye Food Machinery Factory, Guangzhou, Guangdong, China). The formulations are presented in Table 1. In the groups of improved sausages, a partial replacement of pork fat by tremella was performed as followed: TR25 (75% pork fat and 25% tremella); TR50 (50% pork fat and 50% tremella); TR75 (25% pork fat and 75% tremella) and TR100 (0% pork fat and 100% tremella). Each sausage formulation was replicated three times. According to the formulations, the above materials were homogenized for 140 s in a blender (Busch, Marburg, Germany). After the homogeneity, the raw material was put into the casing by sausage stuffer (Shandong Yaobang Equipment Co., Ltd., Linyi, Shandong, China). The sausages were baked at 68 °C for 30 min, braised at 80 °C for 50 min, and fumigated at 50 °C for 150 min by electric heating flue gas furnace (Jiaxing Case stainless steel machinery manufacturing Co., Ltd., Jiaxing, Zhejiang, China). The cooked sausages were cooled in ice-water bath, then put in polythene bags using a vacuum packing machine (Shanghai Yiguang Machinery Co., Ltd., Shanghai, China), and stored at 4 ± 1 °C until analyzing for physicochemical and textural properties.

### 2.3. Proximate Composition

Compositional properties of five groups of sausages were performed according to AOAC [21]. The moisture content was determined by drying at 105 °C for 12 h to calculate the weight loss. The fat content was determined by soxhlet method and solvent extraction method following the AOAC Official Method 930.09. Protein content was measured by Kjeldahl method with a Kjeldahl nitrogen analyzer (KDY-9820, Beijing, China). The samples were burned at 550 °C to determine the ash content. Samples were removed from each treatment and the analysis was carried out at 25 °C in triplicate, and the average was then taken.

### 2.4. Water Activity and pH

The water activity of samples was measured with a water activity meter (Jiangsu Ronghua Instrument Manufacturing Co., Ltd., Jintan, Jiangsu, China). The chopped sausages (10 g) were put into a water activity meter and the data was recorded after 20 min. All determinations were performed at 25 °C in triplicate.

The pH values of sausages were determined by pH meter (Guangzhou Jiayi Precision Instrument Co., Ltd., Guangzhou, Guangdong, China). The samples were obtained in a homogenate (Changzhou Magnetat Instrument Co., Ltd., Guangzhou, Guangdong, China) with 10 g of sausage and 90 mL of distilled water. All determinations were performed at 25 °C in triplicate.

### 2.5. Color

The color of sausages was measured using a colorimeter (CX2064, *L** = 94.52, *a** = −0.86, *b** = 0.68) according to Wang et al. [22]. The *L** (lightness), *a** (redness), and *b** (yellowness) values were measured using standard illuminant D65 light source. And the observation angle was 10°. The aperture of the meter was 14 mm. Four measurements (0° 90° 180° 270°) were taken from each sample surface. The test was repeated three times, and the average was taken. The total color difference (ΔE) and whiteness were calculated as follows [23]:(1)$$\Delta E={\left(\Delta a^{2}+\Delta b^{2}+\Delta L^{2}\right)}^{1/2}$$
(2)$$\mathrm{Whiteness}=100-{\left[{\left(100-L\right)}^{2}+a^{2}+b^{2}\right]}^{1/2}$$
where Δa = *a** − *a** sample, Δb = *b** − *b** sample, ΔL = *L** − *L** sample.

### 2.6. Cooking Loss and Water Holding Capacity (WHC)

Cooking loss was determined with the method of Wang et al. [22]. Each raw sausage was cooked at 80 °C for 50 min. The cooking loss of sausage was obtained by measuring their weight before and after cooking as followed:(3)$$\mathrm{Cooking}\;\mathrm{loss}\left(\%\right)\;=\;\left[\left(m_{1}-m_{2}\right)∕m_{1}\right]\;\times \;100$$
where m_1_ was the weight of raw sausage and m_2_ was the weight of cooked sausage.

The WHC of sausage was obtained with modified methods of Shin et al. [24] and Wang et al. [23]. Approximately 10 g of sausage samples were weighed and placed in a centrifuge tube to centrifuge (H1650R/1850R, Xiangyi Saide Instrument Co., Ltd., Changsha, Hunan, China) for 30 min at 12,000× *g* at 4 °C, and then weighed. The WHC was calculated as followed:(4)$$\mathrm{WHC}\left(\%\right)\;=\;\left(W_{2}∕W_{1}\right)\;\times \;100$$
where W_1_ was the weight of the sausage before centrifugation and W_2_ was the weight of the sausage after centrifugation.

### 2.7. Textural Profile Analysis (TPA)

The texture of cooked sausage was determined using a texture analyzer (TMS-Pro, FTC, Beijing, China). The conditions of texture analysis were as followed: TPA (1000 N), pause time 20 s, force induction 1000 N, probe retraction 30 mm, shape variable 50 s, detection speed 60 mm/min, minimum force 0.8 N. The samples were sliced into pieces of diameter 1.5 cm diameter and 1 cm thickness. All determinations were performed at 25 °C in triplicate. 

### 2.8. Free Amino Acids

Amino acid contents were measured using ninhydrin reagent and separated by cation-exchange chromatography, using an L-8900 automatic amino acid analyzer (Hitachi High-Technologies Corporation, Tokyo, Japan) and a #2622PF column (60 × 4.6 mm) following the method of Jo et al. [25]. The sample (1 g) was hydrolyzed with 10 mL of HCl (6 mol/L) at 110 ± 1 °C for 22 h, cooled to 25 °C, and then filtered. The solvent was removed at 50 °C by rotary evaporator (Zhengzhou Kaixiang instrument equipment Co., Ltd., Zhengzhou, Henan, China). The sodium citrate buffer solution was added to the dried sample, and the mixture was shaken evenly, filtered through 0.22 μm membrane and transferred to the injection bottle as sample determination solution. Samples were measured under the following conditions: AccQ-Tag Ultra C18 column, detection wavelength 260 nm, column temperature 49 °C, sample temperature 20 °C, and flow rate 0.7 mL/min.

### 2.9. Sensory Evaluation

The cooked sausages were evaluated in terms of appearance, texture, flavor, aroma, and overall acceptability. The sensory panel consisted of 30 staffs and graduate students from the Department of Food Science and Engineering at Jilin Agriculture University. Sausages were sliced into pieces of 9 mm thickness and identified with a three-digit random code. In addition, the sensory panel were provided with water and salt-free biscuits to clean their taste buds. Five samples presented at a time. The appearance, texture, flavor, aroma, and overall acceptability were evaluated using a 9-point Hedonic scale (1 = dislike very much, 9 = like very much).

### 2.10. Statistical Analysis

All determinations were designed three times and the values were showed as means ± standard deviations. An analysis of variance was performed on the results measured using SPSS software. The Duncan’s multiple range tests were used to determine the differences among mean values (*p* < 0.05).

## 3. Results

### 3.1. Proximate Composition

The proximate composition expressed in wet base was influenced by the addition of fat and tremella within the formulation of sausages (Table 2). It could be seen that the protein content of the sausage in the control was 12.65%, which was similar to the protein content of the konjac gel with vegetable powders sausage and pork-skin gelatin powder sausage produced by Kim et al. [17] and Lee, Chang, and Hoon [26], respectively. With the increasing of the replacement ratio of tremella the protein content in sausage increased significantly (*p* < 0.05). Compared with the control, the protein content of the replacement group increased by 4.66% to 14.78%, since the protein content of tremella (14%) was higher than that of fat. This result was similar to the protein content of sausage with peanut and linseed oil emulsion gels instead of pork fat by Nacak et al. [27].

The fat content of the sausage in the control was 17.77%, which was similar to the pork sausage in the control reported by Wang et al. [22]. With the increase of tremella, the fat content of sausages in replacement group decreased significantly (*p* < 0.05). Compared with the control, the fat content of the replacement group was reduced by 14.57% to 88.97%, since the fat content in the tremella was much lower than pork fat. Stefanello et al. [28] reported that sun mushroom powder was used as a fat replacer for fresh pork sausage, the fat of sausage was reduced by 4.02% to 5.62%. It showed that the addition of tremella could reduce the fat content of sausage, which was very suitable for people to lose weight, and was a healthy food. Compared with the control, the ash content of the replacement group increased by 4.17% to 14.42%. It might be due to the fact that the mineral content increased with the addition of tremella.

### 3.2. Water Activity and pH

The water activity and pH of different sausage groups were showed in Table 2. Substituting tremella for fat in sausages resulted no significant changes in water activity, which was similar to the research result of Sousa et al. [29], who replaced pork backfat with hydrolyzed collagen in frankfurter-type sausages. With the gradual increasing of fat replacement by tremella, the pH of sausage showed an overall downward trend, probably since the pH of tremella was lower than that of fat. 

### 3.3. Color

Color is an important standard for consumers to accept processed meat [30]. Addition of tremella significantly affected the lightness (*L**-value), redness (*a**-value), yellowness (*b**-value), total color difference (Δ*E**), and whiteness values of sausages (Table 3). With the increase of the proportion of tremella in the sausage, the L and A values of the sausage increased significantly (*p* < 0.05), indicating that the sausage was more shiny and redder. The total color difference of the sausage replaced with tremella was higher than the control. And the more replaced parts, the greater the total color difference. With the proportion of fat substitutes increasing, the whiteness of sausages increased significantly (*p* < 0.05), which might be related to the color of the tremella itself.

### 3.4. Cooking Loss and Water Holding Capacity (WHC)

Table 4 showed the cooking loss and water holding capacity of sausages formulated with tremella. With the increasing ratio of tremella, the cooking loss of sausage generally increased (*p* < 0.05). It might be due to the higher water content of tremella, which evaporated during the cooking process, resulting in more cooking loss. This result was similar to that of Wang adding shiitake mushrooms to replace lean pork in sausages [23]. Compared with the control, the WHC of the replacement group was increased by 1.17% to 9.78%. The reason may be that the WHC of tremella is better than pork fat.

### 3.5. Textural Profile Analysis (TPA)

Better texture was related to the taste and other sensory properties of meat products [31]. The addition of tremella affected the apparent texture profile of sausages (Table 5). The hardness, gumminess, and cohesiveness of the replacement group were significantly lower than those of the control (*p* < 0.05). Since tremella prevented pork myofibril protein from producing gel, resulting in a decrease in the cohesiveness of sausage fillings and gaps appear inside. Compared with the control, the hardness of sausage decreased by 4.58% to 18.52%, the cohesiveness decreased by 20.00% to 29.09%, and the gumminess decreased by 28.31% to 45.32%. However, with the increasing of fat substitutes, the springiness and chewiness of sausages increased significantly (*p* < 0.05). When the replacement ratio of tremella was 100%, the springiness and chewiness of sausages were significantly higher than those of the control, which indicated that the addition of tremella gives the sausage a better taste and makes the sausage have better elasticity and chewiness. This result was similar to Miriam using pork skin and amorphous cellulose to make Bologna-type sausages [32].

### 3.6. Free Amino Acids

Amino acid is an important factor affecting meat quality [25]. The addition of tremella had a significant effect on the free amino acids of sausage samples (Table 6). It could be seen from the table that the essential amino acids of the control sausage were mainly valine, threonine, lysine and leucine; the non-essential amino acids were mainly arginine, proline, glutamic acid and aspartic acid. With the gradual increasing ratio in the replacement of tremella, the content of lysine, isoleucine, proline, and tyrosine in sausage also gradually increased. Compared with the control, the content of lysine, isoleucine, proline, and tyrosine in sausage increased by 16.06% to 47.45%, by 2.5% to 46.25%, by 4.17% to 42.86%, and by 14.58% to 54.17%, respectively. One of the reasons might be that these amino acids were abundant in tremella. Tremella was rich in amino acids [19]. Amino acids have many health benefits. For instance, Lysine could promote the development of human body, enhance immunity and improve the central nervous function [33]. And proline could keep muscles and joint flexible and had the effect of reducing sagging and wrinkling of the skin caused UV exposure and normal aging [34]. In addition, the contents of threonine, methionine, phenylalanine, histidine, and cysteine were similar to those of the control, which showed that the addition of tremella had little effect on them.

### 3.7. Sensory Evaluation

Sensory evaluations of each group of sausage were performed in Figure 1. Both the control and TR100 were lower in flavor than the other substitute groups, which might be caused by the single flavor in the sausage. In terms of appearance, texture and flavor, the score of TR100 was lower than that of other groups. It might be caused by the soft texture of the sausage and the gaps in the cut surface when the tremella completely replaces the fat. TR25 was close to the control in terms of texture, aroma, flavor and appearance. It might since the amount of replacement of tremella was less and there was no significant difference compared with the control. TR75 had higher scores in flavor and aroma than control. Since sausages not only increase the unique aroma of tremella, but also reduce the greasy flavor of sausages, which were deeply loved by the group members. The lower texture score of TR75 was due to the fact that the sausage was softer and the filling was looser. However, all of these samples were judged acceptable (>5) by team members. Through the above test, it could be seen from the above tests that TR75 has preferable characteristics in all aspects.

## 4. Conclusions

This study showed that the substitution of pork fat with tremella in the sausage formulation could be a strategy with great promise to improve nutritional quality of the sausage. The fat content of sausage with tremella was decreased by 14.57% to 88.97%. The content of protein and ash of sausage were increased by 4.66% to 14.78% and 4.17% to 14.42%, respectively. Furthermore, the use of tremella improved the lightness, redness, water holding capacity, and textural profile analysis of the sausage. Moreover, some free amino acid increased, including lysine increased by 16.06% to 47.45%, isoleucine increased by 2.5% to 46.25%, proline increased by 4.17% to 42.86%, and tyrosine increased by 14.58% to 54.17%. From a sensory point of view, the best formulation for replacement proportion of tremella was 75%, which not only enhanced the aroma and flavor of the sausage, but also gave it a unique aroma. Based on these results, we are confident that tremella could be a promising ingredient for enhancing sausage flavor and overall quality.

## Figures and Tables

**Figure 1 foods-10-02167-f001:**
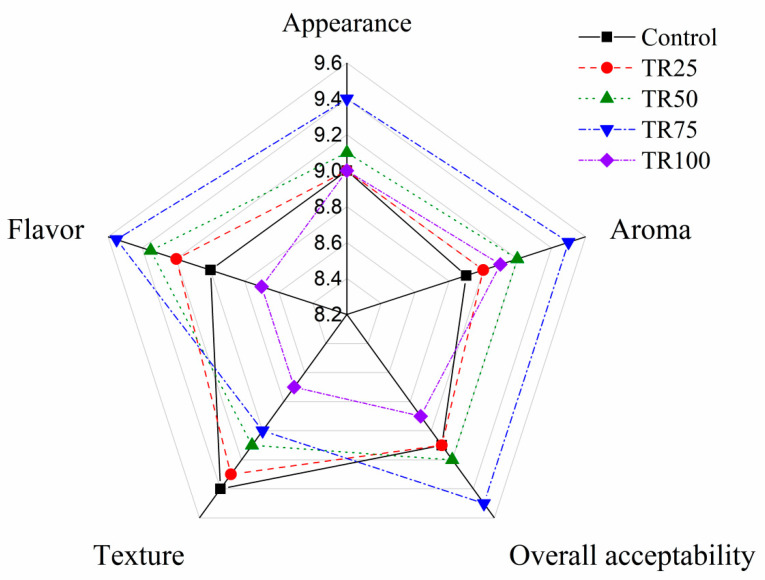
Sensory evaluations of each group of sausages with pork fat replaced by tremella.

**Table 1 foods-10-02167-t001:** Formulations of sausages with pork fat replaced by tremella.

Formulations (%)	Control	TR25	TR50	TR75	TR100
Pork lean meat	49	49	49	49	49
pork fat	21	15.75	10.5	5.25	0
tremella	0	5.25	10.5	15.75	21
Salt	1.5	1.5	1.5	1.5	1.5
Sugar	1	1	1	1	1
White pepper	0.2	0.2	0.2	0.2	0.2
Carrageenan	0.3	0.3	0.3	0.3	0.3
Isolated soy protein	2.8	2.8	2.8	2.8	2.8
Dry starch	4.2	4.2	4.2	4.2	4.2
Ice	20	20	20	20	20
Total	100	100	100	100	100

**Table 2 foods-10-02167-t002:** The proximate composition expressed in wet base, water activity and pH of sausages with pork fat replaced by tremella.

Parameters	Control	TR25	TR50	TR75	TR100
Protein (%)	12.65 ± 0.14 ^a^	13.24 ± 0.16 ^b^	13.63 ± 0.04 ^c^	14.16 ± 0.03 ^d^	14.52 ± 0.38 ^e^
Fat (%)	17.77 ± 0.65 ^e^	15.18 ± 0.5 ^d^	8.40 ± 0.08 ^c^	6.17 ± 0.13 ^b^	1.96 ± 0.27 ^a^
Ash (%)	3.12 ± 0.0 ^a^	3.25 ± 0.02 ^b^	3.36 ± 0.04 ^c^	3.52 ± 0.01 ^d^	3.57 ± 0.01 ^e^
Moisture (%)	46.63 ± 0.02 ^a^	58.01 ± 0.0 ^b^	67.08 ± 0.01 ^c^	69.08 ± 0.02 ^d^	73.29 ± 0.01 ^e^
Aw	0.99 ± 0.0 ^a^	0.98 ± 0.00 ^a^	0.99 ± 0.00 ^a^	0.99 ± 0.00 ^a^	0.99 ± 0.00 ^a^
pH	6.46 ± 0.05 ^b^	6.45 ± 0.02 ^c^	6.33 ± 0.03 ^b^	6.27 ± 0.01 ^a^	6.26 ± 0.01 ^a^

^a–e^ Means within the same row with different letters differ significantly among the treatments (*p* < 0.05). Control, TR25, TR50, TR75, and TR100 were 0%, 25%, 50%, 75%, and 100% replacement of pork fat by tremella, respectively.

**Table 3 foods-10-02167-t003:** Color parameters of sausages with pork fat replaced by tremella.

Parameters	Control	TR25	TR50	TR75	TR100
*L**	54.37 ± 0.06 ^a^	54.46 ± 0.12 ^a^	55.24 ± 0.10 ^b^	55.91 ± 0.03 ^c^	56.72 ± 0.12 ^d^
*a**	16.06 ± 0.15 ^a^	16.21 ± 0.17 ^ab^	16.36 ± 0.01 ^b^	16.44 ± 0.09 ^b^	17.03 ± 0.07 ^c^
*b**	15.08 ± 0.16 ^d^	14.65 ± 0.05 ^c^	14.02 ± 0.11 ^b^	13.79 ± 0.26 ^b^	12.94 ± 0.04 ^a^
Δ*E**	45.76 ± 0.37 ^d^	45.73 ± 0.06 ^d^	44.92 ± 0.12 ^c^	44.29 ± 0.07 ^b^	43.58 ± 0.14 ^a^
whiteness	49.44 ± 0.33 ^a^	49.49 ± 0.07 ^a^	50.32 ± 0.12 ^b^	50.97 ± 0.07 ^c^	51.72 ± 0.14 ^d^

^a–d^ Means within the same row with different letters differ significantly among the treatments (*p* < 0.05). Control, TR25, TR50, TR75 and TR100 were 0%, 25%, 50%, 75% and 100% replacement of pork fat by tremella, respectively.

**Table 4 foods-10-02167-t004:** Cooking loss and water holding capacity of sausages with pork fat replaced by tremella.

Parameters	Control	TR25	TR50	TR75	TR100
Cooking loss (%)	5.13 ± 0.00 ^a^	12.07 ± 0.04 ^c^	12.54 ± 0.05 ^d^	12.00 ± 0.03 ^b^	14.79 ± 0.15 ^e^
WHC	82.08 ± 0.10 ^a^	83.04 ± 0.13 ^a^	85.79 ± 0.11 ^b^	88.39 ± 0.08 ^bc^	90.11 ± 0.21 ^c^

^a–e^ Means within the same row with different letters differ significantly among the treatments (*p* < 0.05). Control, TR25, TR50, TR75 and TR100 were 0%, 25%, 50%, 75%, and 100% replacement of pork fat by tremella, respectively.

**Table 5 foods-10-02167-t005:** Texture profile analysis of sausages with pork fat replaced by tremella.

Parameters	Control	TR25	TR50	TR75	TR100
Hardness (N)	155.63 ± 10.14 ^c^	148.50 ± 6.02 ^bc^	140.60 ± 4.46 ^abc^	138.90 ± 6.40 ^ab^	126.80 ± 5.36 ^a^
Cohesiveness	0.55 ± 0.03 ^b^	0.44 ± 0.04 ^a^	0.43 ± 0.02 ^a^	0.41 ± 0.03 ^a^	0.39 ± 0.02 ^a^
Springiness	3.50 ± 0.22 ^a^	4.35 ± 0.14 ^b^	4.60 ± 0.12 ^bc^	4.86 ± 0.22 ^c^	5.47 ± 0.24 ^d^
Gumminess (N)	85.23 ± 3.31 ^c^	61.10 ± 1.28 ^b^	56.10 ± 3.42 ^ab^	51.30 ± 2.60 ^ab^	46.60 ± 2.50 ^a^
Chewiness (N)	248.49 ± 8.06 ^a^	257.08 ± 7.35 ^a^	296.79 ± 10.51 ^b^	316.75 ± 6.32 ^b^	371.16 ± 5.62 ^c^

^a–d^ Means within the same row with different letters differ significantly among the treatments (*p* < 0.05). Control, TR25, TR50, TR75 and TR100 were 0%, 25%, 50%, 75%, and 100% replacement of pork fat by tremella, respectively.

**Table 6 foods-10-02167-t006:** Free amino acids (expressed as mg/100 g of sausage) profile of sausages with pork fat replaced by tremella.

Amino Acid	Control	TR25	TR50	TR75	TR100
**Essential**					
Val	1.06 ± 0.02 ^e^	0.99 ± 0.00 ^d^	0.85 ± 0.01 ^c^	0.73 ± 0.00 ^b^	0.68 ± 0.05 ^a^
Thr	1.14 ± 0.00 ^b^	1.11 ± 0.00 ^b^	0.98 ± 0.01 ^b^	0.82 ± 0.02 ^a^	0.79 ± 0.00 ^a^
Lys	1.37 ± 0.00 ^a^	1.59 ± 0.00 ^b^	1.92 ± 0.00 ^c^	1.98 ± 0.00 ^d^	2.02 ± 0.06 ^e^
Met	0.08 ± 0.00 ^a^	0.09 ± 0.00 ^a^	0.11 ± 0.01 ^ab^	0.11 ± 0.00 ^ab^	0.16 ± 0.01 ^b^
Ile	0.80 ± 0.01 ^a^	0.82 ± 0.00 ^a^	1.02 ± 0.01 ^b^	1.12 ± 0.00 ^c^	1.17 ± 0.00 ^d^
Leu	1.89 ± 0.01 ^d^	1.79 ± 0.03 ^c^	1.75 ± 0.00 ^c^	1.26 ± 0.01 ^a^	1.43 ± 0.03 ^b^
Phe	0.99 ± 0.01 ^c^	0.92 ± 0.00 ^b^	0.89 ± 0.00 ^a^	0.90 ± 0.03 ^a^	0.95 ± 0.00 ^b^
His	0.79 ± 0.02 ^b^	0.79 ± 0.06 ^b^	0.72 ± 0.01 ^a^	0.70 ± 0.00 ^a^	0.78 ± 0.00 ^b^
Val	1.06 ± 0.02 ^e^	0.99 ± 0.00 ^d^	0.85 ± 0.01 ^c^	0.73 ± 0.00 ^b^	0.68 ± 0.05 ^a^
**Non-Essential**					
Ser	0.98 ± 0.00 ^c^	0.88 ± 0.00 ^a^	0.92 ± 0.00 ^b^	0.85 ± 0.01 ^a^	0.93 ± 0.01 ^b^
Arg	1.42 ± 0.04 ^e^	1.38 ± 0.00 ^d^	1.29 ± 0.00 ^c^	1.14 ± 0.00 ^b^	0.93 ± 0.03 ^a^
Gly	1.11 ± 0.00 ^d^	1.12 ± 0.00 ^d^	0.83 ± 0.00 ^b^	0.89 ± 0.00 ^c^	0.74 ± 0.00 ^a^
ASP	2.62 ± 0.00 ^e^	2.36 ± 0.03 ^d^	2.27 ± 0.06 ^c^	1.89 ± 0.05 ^b^	1.73 ± 0.01 ^a^
Glu	4.02 ± 0.03 ^d^	3.84 ± 0.05 ^c^	3.79 ± 0.03 ^c^	3.58 ± 0.01 ^b^	2.78 ± 0.05 ^a^
Ala	1.30 ± 0.00 ^d^	0.88 ± 0.04 ^a^	1.18 ± 0.00 ^c^	0.93 ± 0.00 ^b^	1.19 ± 0.03 ^c^
Pro	1.68 ± 0.03 ^a^	1.75 ± 0.00 ^b^	2.07 ± 0.02 ^c^	2.09 ± 0.03 ^c^	2.40 ± 0.26 ^d^
Cys	0.06 ± 0.00 ^a^	0.03 ± 0.00 ^a^	0.05 ± 0.00 ^a^	0.03 ± 0.01 ^a^	0.05 ± 0.33 ^a^
Tyr	0.48 ± 0.01 ^a^	0.55 ± 0.00 ^b^	0.65 ± 0.00 ^c^	0.68 ± 0.00 ^c^	0.74 ± 0.01 ^d^

^a–e^ Means within the same row with different letters differ significantly among the treatments (*p* < 0.05). Control, TR25, TR50, TR75, and TR100 were 0%, 25%, 50%, 75%, and 100% replacement of pork fat by tremella, respectively.

## Data Availability

The data presented in this study are available in the article.

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
