# Peer review of "Use of Tremella as Fat Substitute for the Enhancement of Physicochemical and Sensory Profiles of Pork Sausage"

_foods, 2021, doi:10.3390/foods10092167_

Round 1

Reviewer 1 Report

Please find attached my comments for consideration regarding the presentation of this material. Authors have done a nice job but the presentation would be enhanced with some added clarity. 

Reviewer 2 Report

The manuscript describes the effect of the partial or total substitution of fat in sausages for the fungus Tremella. The authors prepared four different sausages with different percentages of substituted fat and they report several physicochemical and technological features of the sausages (composition, colour, water activity, etc.) as well as a sensory evaluation. The research itself is very interesting and the potential of Tremella as a fat replacer seems to be very promising.

General comments

English language should be checked carefully throughout the manuscript.

Subsection 2.1. Materials. The authors did not specify what species of tremella they used for this study. Moreover, the proximate analysis of this ingredient is essential to better understand how its addition to the sausage affects the final composition of the product. This information should be included in the manuscript, either retrieved from the literature or, if possible, from a proximate analysis carried out by the authors under the same conditions as the samples. I also recommend including some information about why the composition of tremella makes it suitable as a fat replacer. What components provide the texture and mouthfeel appropriate as a fat replacer?

Subsection 2.9. Sensory evaluation. Some details about the sample presentation should be added. Were the samples presented monadically, i.e. one at a time? Type of random code used for the samples? Three digits? Was there any type of statistical analysis performed to the results from the sensory evaluation? Were the differences reported significant?

Table 2. Is the proximate composition expressed in dry or wet base? Please specify.

Round 2

Reviewer 1 Report

Would suggest to authors that they include p-values within the abstract where statements of characteristics for the sausages have been increased, improved, or best. Abstract is the first portion of the article that readers will see and accuracy through interpretation of results in the abstract is beneficial to readership. 
